**Data Availability Statement:** All data files are available from Mendeley Data under the following DOI link: http://dx.doi.org/10.17632/cwj76cbvc9.1.

# Stool metabolome-microbiota evaluation among children and adolescents with obesity, overweight, and normal-weight using 1H NMR and 16S rRNA gene profiling

José Diógenes Jaimes[1], Andrea Slavíčková[1‡], Jakub Hurych[2‡], Ondřej Cinek[2,3‡], Ben Nichols[4], Lucie Vodolánová[2], Karel Černý[5], Jaroslav Havlík[1]*

1 Department of Food Science, Czech University of Life Sciences Prague, Prague, Czech Republic, 2 Department of Medical Microbiology, 2nd Faculty of Medicine, Charles University, Motol University Hospital, Prague, Czech Republic, 3 Department of Paediatrics, 2nd Faculty of Medicine, Charles University, Motol University Hospital, Prague, Czech Republic, 4 Human Nutrition, School of Medicine, Dentistry & Nursing, College of Medical, Veterinary and Life Sciences, University of Glasgow, Glasgow Royal Infirmary, Glasgow, United Kingdom, 5 Olivova Children's Medical Institution, Říčany, Czech Republic

☯ These authors contributed equally to this work.
‡ These authors also contributed equally to this work.
* havlik@af.czu.cz

## Abstract

Characterization of metabolites and microbiota composition from human stool provides powerful insight into the molecular phenotypic difference between subjects with normal weight and those with overweight/obesity. The aim of this study was to identify potential metabolic and bacterial signatures from stool that distinguish the overweight/obesity state in children/adolescents. Using 1H NMR spectral analysis and 16S rRNA gene profiling, the fecal metabolic profile and bacterial composition from 52 children aged 7 to 16 was evaluated. The children were classified into three groups (16 with normal-weight, 17 with overweight, 19 with obesity). The metabolomic analysis identified four metabolites that were significantly different ($p < 0.05$) among the study groups based on one-way ANOVA testing: arabinose, butyrate, galactose, and trimethylamine. Significantly different ($p < 0.01$) genus-level taxa based on edgeR differential abundance tests were genus *Escherichia* and *Tyzzerella* subgroup 3. No significant difference in alpha-diversity was detected among the three study groups, and no significant correlations were found between the significant taxa and metabolites. The findings support the hypothesis of increased energy harvest in obesity by human gut bacteria through the growing observation of increased fecal butyrate in children with overweight/obesity, as well as an increase of certain monosaccharides in the stool. Also supported is the increase of trimethylamine as an indicator of an unhealthy state.

The items included there are: 1) S1 Table 1 (as referenced in the manuscript) containing the individual participant's gender, age, BMI, BMI z-score, as well as the kilocalorie and macronutrient daily percentage composition one and two days prior to sampling; 2) unrarefied source data for the 16S rRNA gene sequencing analysis; 3) Metabolite concentrations in mg/g derived from Chenomx NMR Suite version 7.5 for each of the 52 study participants; and 4) the 1H NMR spectra for the 52 study participants.

**Funding:** Funding for this research was provided by the Ministry of Education, Youth and Sports of the Czech Republic, research grants INTER-COST LTC19008 and METROFOOD-CZ research infrastructure project LM2018100, both awarded to JH. The funder website is https://www.msmt.cz/. The funder had no role in study design, data collection and analysis, decision to publish, or preparation of the manuscript.

**Competing interests:** The authors have declared that no competing interests exist.

## Introduction

The proportion of children and adolescents aged 5 to 19 considered overweight has risen globally from approximately 1 in 10 in the mid-1970s to about to about 1 in 5 in 2016 [1]. There is strong evidence of a close relationship between childhood overweight/obesity and multiple comorbidities which, collectively, reduce life expectancy and increase mortality. This has become an emerging public health problem that has attracted the wide attention of researchers [2–4]. Identifying potential biomarkers in pediatric populations via metabolomics and 16S rDNA profiling can provide an opportunity not only to identify these conditions, but to find potential prevention and treatment approaches.

Despite inter-individual differences, approaches using 16S rRNA gene amplicon sequencing (16S rDNA profiling) of fecal samples have shown differing gut bacterial composition between children with obesity and those without [5]. For example, at the phylum level, a high Firmicutes/ Bacteroidetes ratio (decrease in Bacteroidetes, increase in Firmicutes) has been associated with obesity [6–8], and even specific strains such as *F. prausnitzii* have been positively correlated with BMI z-score [8]. Nevertheless, there is inconsistency in these observations and knowledge about the specific gut microbiota members relevant to the characterization of overweight/obesity remain elusive [6, 9, 10]. Furthermore, microbiome studies have thus far tended to focus on adult populations; consequently, compositional and functional differences between children and adult cohorts have not been reported [11].

Although powerful, 16S rDNA profiling falls short in telling us about the functional activities of the gut microbiota [12]. Another omics approach that helps fill this void is metabolomics. Fecal metabolomics in particular reports on the interaction between host, diet, and the microbiota, thus complementing 16S profiling by providing a functional readout of the microbiota [12, 13] and thus providing a characterization of the molecular phenotype [14–16]. Metabolomic studies have observed certain metabolic patterns and signatures as potential biomarkers of obesity. For example, studies using various biofluids have observed that an increased level of branched-chain amino acids (valine, leucine, isoleucine) and of aromatic amino acids (tyrosine, tryptophan, phenylalanine, methionine) appear to characterize the presence, and, in some cases, the propensity for obesity [2, 17, 18]. Nevertheless, further research is necessary to test whether proposed biomarker metabolites can be considered an established and specific metabolic signature [17]. Furthermore, it is important to differentiate metabolomic profile differences between children and adults; for example, one striking observed difference in childhood obesity in contrast to adult obesity is that impairment of fasting glucose levels is usually absent and, if present, it is a delayed finding [2]. A recent example integrating 16S rDNA profiling and metabolomics observed that 67.7% of the fecal metabolome variance was explained by the gut microbiota composition [12] and it has been widely observed that changes in metabolite levels are often associated with the microbiota [2, 5, 17, 19]. This high degree of association between the microbiota and the fecal metabolome makes integration of these two omics technologies a powerful investigative strategy.

This study integrates these two omics approaches. Using stool samples as the analytical matrix and an untargeted approach, we aimed to uncover potential metabolomic and gut bacterial biomarkers of childhood overweight/obesity in a group of 52 Czech children/adolescents aged 7 to 16 years. The analytical platforms used were $^1$H NMR to evaluate the metabolomic profile and 16S rRNA gene sequencing to assess the gut bacteria composition. The significant results from each were then correlated to better define metabolome-gut bacteria relations.

## Methods

### Study participants

This is an observational study to characterize differences in the stool metabolome and bacteriome among 52 Czech youth (28 females, 24 males) aged 7 to 16 years classified into three comparison groups (normal-weight, overweight, and obese). The study was performed according to the Declaration of Helsinki and was approved by the Ethical Committee of the University Hospital Královské Vinohrady, reference number LEK-VP / 01/0/2018. All participants and their parents agreed to participate by signing appropriate written informed consents. These written informed consents were provided by the parents for the participation of their children in the study. The inpatient study took place during an eight-week period between late July to late September 2018.

Participant recruitment was carried out by the Olivova Children's Medical Institution (Olivova Dětská Léčebna) in Říčany, Czech Republic, from their then-present patient population. Recruitment criteria consisted of youth who were between 6 and 18 years of age, had no antibiotic intake for the past three months, were not currently taking any medication, and were physically healthy to take part in a physical activity program led by physiotherapists focused on aerobic activity, strengthening, and stretching twice a day on weekdays. The subjects considered overweight and obese were under treatment/rehabilitation of their overweight/obesity condition through diet and physical activity. The control group (normal weight) were youth with either a respiratory disease (chronic upper and lower respiratory cataracts, bronchial asthma, allergies) or with an orthopedic diagnosis (scoliosis, poor posture, patients after surgery). To minimize the potential of these conditions being confounding, those with a respiratory disease had to be in remission (not in an acute stage) whose only treatment was climatotherapy and respiratory physiotherapy. Those with an orthopedic diagnosis were also only receiving physiotherapy. Out of 121 identified youth who met these conditions, 52 of them ultimately agreed (along with their parents) and/or completed the entire study process. Once in the study, all participants were interned at Olivova for an eight-week period during which they underwent a similar physical activity regime and all received a similar diet (same ingredients, same dishes). The portion size for each participant was based on their age, gender, and weight. Those classified as overweight/obese received 30% less kilocalories than recommended for their age and gender, thus placing them on a caloric restriction. It is important to note that the diet was planned by a clinical dietitian, the meals and snacks were prepared by the Olivova cafeteria kitchen, and the composition of each meal (kilocalories, macronutrient composition) was known as determined by the disaggregated ingredients of each meal through the use of the Nutriservis Profi software (https://nutriservis.cz), a database of approximately 5000 ingredients, including over 900 Czech ones. There were three main meals provided (breakfast, lunch, dinner) plus two snacks throughout the day. The meals were based on recipes and foods typically eaten in a standard Czech school and home diet, thus, except for portion control, the meals did not represent an adjustment for most participants. Although the participants and their families agree to comply with the diet, potential leftover food and/or additional intake of other items outside the provided meals could not be accounted for. On average, the meals were composed of approximately 17% protein, 28% fat and 55% carbohydrates. This is a study period average, thus daily percentages were different. Participants were not all sampled at the same time, but throughout the study period with the earliest collections taking place after at least one week of habituation to the prescribed diet. As a result of the selection criteria, a similar physical activity regime, a homogenous diet among the participants, and the same ethnic and geographic background of the participants, the effects of cofounding variables were minimized.

**Table 1. Characteristics of the participants per study group (N, OW, OB).**

|  | N | OW | OB | p-value[a] |
|---|---|---|---|---|
| n | 16 | 17 | 19 | - |
| Gender | M = 7, F = 9 | M = 6, F = 11 | M = 11, F = 8 | - |
| Age (years) | 11.06 (± 2.46) | 11.47 (± 2.24) | 10.47 (± 2.37) | 0.45 |
| BMI (kg/m$^2$) | 18.05 (± 2.45) | 23.93 (± 3.10) | 30.17 (± 4.28) | < 0.01 |
| BMI z-score | 0.07 (± 0.80) | 1.53 (± 0.29) | 2.37 (± 0.23) | < 0.01 |
| % Carb. D-1 | 64.06 (± 5.09) | 63.33 (± 3.93) | 64.02 (± 4.32) | 0.87 |
| % Prot. D-1 | 18.51 (± 1.70) | 19.55 (± 1.68) | 19.12 (± 2.07) | 0.29 |
| % Fat D-1 | 17.43 (± 4.93) | 17.11 (± 3.12) | 16.86 (± 3.78) | 0.92 |
| % Carb. D-2 | 66.46 (± 3.18) | 65.13 (± 6.64) | 64.96 (± 5.89) | 0.69 |
| % Prot. D-2 | 18.29 (± 4.26) | 21.00 (± 3.83) | 19.43 (± 2.18) | 0.10 |
| % Fat D-2 | 15.25 (± 3.24) | 13.88 (± 4.87) | 15.61 (± 4.66) | 0.48 |
| kcal D-1 | 2224.80 (± 309.28) | 1574.96 (± 154.58) | 1565.42 (± 137.60) | < 0.01 |
| kcal D-2 | 2066.15 (± 322.74) | 1533.68 (± 129.93) | 1553.61 (± 77.66) | < 0.01 |

M = male, F = female; values reported as mean (± standard deviation); D-1 = one day prior to sampling; D-2 = two days prior to sampling.

[a]. p-value based on one-way ANOVA test.

The participants' body mass index (BMI) standard deviation score (z-score) was derived from age-specific and sex-specific parameters from the Czech National Institute of Public Health [20]. Based on World Health Organization's guidelines [21], these data were used to classify them into three categories: with obesity (OB) Z-scores > 2.00, with overweight (OW) Z-scores > 1.00, and with normal (N) Z-scores ≤ 1.00 and ≥ -2.00. In total, 16 classified with normal-weight (N), 17 with overweight (OW), and 19 with obesity (OB). Table 1 displays, per each of the three study groups, the size, gender, as well as the mean (± standard deviation) for the age, body mass index (BMI), BMI z-score, and the dietary percent macronutrient composition and kilocalorie content one day prior (D-1) and two days prior (D-2) the date of sampling. S1 Table provides similar data per each participant.

## Sample collection

Stool sample collection was carried out by the child/adolescents themselves after proper instruction of the use of a disposable kit that consisted of a paper collection surface from which approximately 1 g of stool was collected with a plastic spoon and deposited into a plastic vial. The sample was then given to a nurse and stored at—20 ˚C until it was transported to the analysis lab where it was stored at -80 ˚C until the time of the analysis.

After thawing, three stool aliquots of approximately 200 mg were placed into three 1.5 mL microcentrifuge tubes. One aliquot was lyophilized to estimate the water content. No water content was reported for two N samples, and for three OB samples. These values were then used to normalize metabolite concentrations to water content as described in the statistics section. The other two aliquots were used for the NMR (metabolomic) and 16S rDNA sequencing analysis. Metabolomic analysis was applied to all 52 samples while 16S rDNA profiling was carried out in 47 samples due to insufficient sample amount (2 from the N and 3 from the OW group).

## Metabolomic analysis

**NMR sample preparation and processing.** After thawing the aliquot from the 1.5 mL microcentrifuge tube, 600 μL of ultrapure water was added. This was then vortexed (3000 rpm,

10sec) and centrifuged (17000 x*g*, 10 min) using a fixed angle rotor. The resulting supernatant (540 μL) was transferred to another 1.5 mL microcentrifuge tube, and 60 μL of phosphate buffered saline (PBS, 1.5 M $K_2HPO_4$ / 1.5 M $NaH_2PO_4$, 5 mM 3-(trimethylsilyl)-2,2,3,3-tetradeuteropropionic acid (TSP) + $D_2O$, 0.2% $NaN_3$, pH 7.4) solution was added. The sample was then centrifuged (17000 x*g*, 10 min) using a fixed angle rotor. The resulting supernatant (500 to 550 μL) was transferred to a 5 mm NMR tube and introduced into an NMR spectrometer for analysis.

**NMR spectroscopy.** [1]H NMR spectra were recorded on a 500.23 MHz Bruker Avance III spectrometer at a temperature of 298 K, equipped with a BBFO SmartProbe™ with Z-axis gradients and a 24 slot autosampler (Bruker Biospin, Germany). A standard Bruker noesypr1d (90˚-t1-90˚-dmix-90˚-FID) sequence was used to suppress signals from water molecules, where t1 is a 4 μs delay time and dmix is the mixing time (0.1 s). Acquisition parameters for the spectra were 128 scans, a 16 ppm spectral width collected into 32K data points, an acquisition time of 4 s, and an interscan relaxation delay of 5 s. Automatic routine including tuning, 3D shimming, 90˚pulse calibration and automatic receiver gain setting was run prior to each sample.

**NMR data processing and analysis.** The Free Induction Decay (FID) obtained were zero-filled to 64 k, Fourier-transformed, manually phased, and baseline corrected and referenced to TSP:0 ppm using TopSpin 3.1 software (Bruker Biospin, Rheinstetten, Germany).

Multivariate analysis (MVA) was carried out via a chemometric approach. The spectra were further manually phased and baseline corrected manually using Whittaker smoother algorithm in MestreNova NMR Suite software package (Ver. 6.0.2, Mestrelab Research, S.L., Spain). Spectra between δ 9.0–0.0 ppm (excluding the residual water region, δ 5.1–4.6 ppm) were reduced into consecutive, non-overlapping bins (buckets) of equal 0.04 ppm widths. Bins were integrated and normalized based on the total sum of the spectral integral. Unsupervised principal component analysis (PCA), and supervised partial least squares discriminant analysis (PLS-DA) were applied to the normalized bins using MetaboAnalyst 3.0 (http://www.metaboanalyst.ca) [22, 23] under the following parameters: no data filtering, sum normalization, no data transformation, pareto scaling. The PLS-DA model was evaluated using a 10-fold cross-validation.

Univariate analysis (UVA) was carried out using a quantitative (deconvolution) approach. Using Chenomx NMR Suite (version 7.5, Chenomx Inc., Edmonton, Canada), fourier-transformed spectra were subject to line broadening of 0.3 Hz, followed by further phase and baseline manual correction. Metabolites were identified via the Chenomx Profiler library, the Human Metabolome Database (http://www.hmdb.ca), and the literature. The metabolite concentrations in mg/dL from Chenomx were adjusted for the sample dilution. Using the individual sample wet weight (g) the concentrations were converted to mg/g, then normalized by the mean water content of the entire data set, and finally Log2 transformed to prevent the dominance of higher abundance metabolites, to decrease the skewness of the data, and to approximate a more normal distribution. Under the hypothesis that there was no significant difference among the three study groups, a one-way analysis of variance (one-way ANOVA) was applied ($p < 0.05$, two-tailed) to the Log2 transformed concentrations. The test accounted for Levene's test for equality of variances and used Tukey's HSD as a post-hoc procedure. For comparison, the false discovery rate (FDR) was also evaluated. Afterwards, Log 2 transformed concentrations of the resulting significant metabolites, after removal of outliers using Tukey's method (above and below 1.5*IQR), were evaluated for the presence and direction of a linear relationship between each metabolite pair and with the z-score through a Pearson correlation ($p < 0.05$). These analyses were carried using R statistical software version 3.6.3. [24].

### 16S rRNA gene profiling

**DNA extraction and qPCR amplification.** DNA for 16S rRNA sequencing was extracted from approximately 50–100 mg of unprocessed stool samples using the DNeasy PowerSoil Kit (Qiagen, Germany) per manufacturer's instructions. Extraction's control was performed by qPCR amplification targeting the V4 region of the 16S rRNA gene.

**Library preparation and 16S rDNA sequencing.** Samples were sequenced in duplicates in a single run. The V4 region of the 16S rDNA gene was amplified using tagged primers by Schloss et al. [25]. using the AccuPrime polymerase blend (Invitrogen, USA). The thermal protocol of the PCR reaction was composed of an initial denaturation at 95˚C for 5 minutes, followed by 30 cycles of 1) denaturation at 95˚C for 15 seconds, 2) primer annealing at 55˚C for 30 seconds and 3) elongation at 68˚C for 1 minute using slow amplification ramp of 1 ˚C per second to reduce chimera formation. A mock community, a mixture of known microbial DNA, was processed along with the research samples. The bacterial mock community was an in-house mixture comprising genomic DNA extracted from cultures of following bacteria, mixed in uneven ratios and frozen in suitably sized aliquots: *Actinomyces odontolyticus*, *Burkholderia cepacia*, *Clostridioides difficile*, *Enterococcus faecalis*, *Escherichia coli*, *Listeria monocytogenes*, *Prevotella denticola*, *Pseudomonas aeruginosa*, *Staphylococcus aureus*, *Staphylococcus epidermidis*, *Streptococcus agalactiae*, *Streptococcus pneumoniae*. The correct identification of the genera (or of species, wherever the V4 region is discriminative) was checked upon completion of the sequencing run. Amplicon size was checked by agarose gel electrophoresis. Amplified libraries of the 16S rDNA gene were purified with Ampure magnetic beads on a Biomek robot (both Beckman Coulter, USA). Purified libraries were then equalized, and pooled. Equalization was based on quantification by a real-time PCR assay using the KAPA library quantification kit (Kapa Biosystems, USA). Data from the qPCR machine were processed by a computer script calculating dilution ratios, and the equalization and pooling was run on a Biomek robot (Beckman Coulter, USA). The final concentration of the pools of 16S rRNA libraries was measured by Qubit dsDNA high-sensitivity assay (Thermo Fisher Scientific, USA). Sequencing was performed on a MiSeq instrument (Illumina, USA) with the sequencing kit for 2x 250 base pairs (Ilumina, USA).

**16S rDNA data analysis.** The ensuing demultiplexed sequencing reads were first trimmed and filtered by quality, dereplicated to remove redundancy, error rates were estimated, and true sequences inferred from the pooled sequencing reads of the whole run. Then the read pairs were merged, chimeras removed, and amplicon sequence variants (i.e. operational taxonomic units) tabulated by samples. Finally, taxonomic assignment was done using the Silva database version 132. These steps were performed using the DADA2 package [26]. The phylogenetic tree was constructed by the neighbour-joining method followed by generalized time-reversible distances with gamma rate variation implemented in the *phangorn* package [27]. Sequences classified as chloroplasts, archaea, or cyanobacteria were removed. Subsequently, the data were converted into a *phyloseq* object [28] and analyzed.

Alpha-diversity was compared among the N, OW, and OB groups using the Chao1, Simpson, Shannon, and ACE indices. Comparisons were carried out under the null assumption that there was no significant difference (p < 0.05) among the three groups. These were carried out at the phylum, family, and genus levels via MicrobiomeAnalyst (https://www.microbiomeanalyst.ca) [29, 30] and *phyloseq* by using the corresponding taxa total abundance after cumulative sum scaling. Due to previously observed association of a high Firmicutes/Bacteroidetes (F/B) ratio with obesity [6–8], the Ln-transformed F/B ratio derived from the relative abundance of these two phyla was tested under the hypothesis that there was no significant difference among the three study groups via a one-way ANOVA (p < 0.05) with

Tukey's HSD as a post-hoc procedure using R statistical software version 3.6.3. [24]. For the F/B ratio, samples 19 (N), 36 (OW), and 29 (OB) were excluded due to extreme values ($\geq 1.5^*$IQR) in the Bacteroidetes counts.

Due to the controversial nature of differential abundance analysis in microbiome research [31, 32] and that no statistical method can fully capture biological phenomena, differential abundance testing was carried out using two techniques: 1) analysis of composition of micro-biomes (ANCOM), which has been shown to provide lower false discovery rate (FDR) than comparable methods [32, 33]; and 2) edgeR, which has also displayed relatively lower FDR (although higher than ANCOM) and has been recommended for overall performance and smaller data sets [33–35]. ANCOM was applied to the total abundance table using a signifi-cance threshold of $\geq 0.8$ and was carried out under the null assumption that among the three different groups there was no significant difference in the relative abundance proportion between each taxon pair at a specific taxonomic level. It was carried out at five taxonomic lev-els (genus to phylum) using the script ANCOM v2.1 [36] in R statistical software version 3.6.3. [24]. EdgeR (significance $p \leq 0.01$ after FDR correction) was carried out under the null hypothesis that taxa were not differentially abundant among the three study groups using MicrobiomeAnalyst under the following parameters: at least 25% of values having a read count of 4 or greater, a variance $> 10\%$ by IQR throughout the experiment, and cumulative sum scaling without rarefaction or transformation.

## Correlation of metabolomic and 16S rDNA analyses

Resulting significant genera from the differential abundance analysis were evaluated for the presence and direction of a relationship among themselves and with the significant metabolite concentrations and the z-score through a correlation test. This was also done globally for all the identified metabolites and genera. The Spearman correlation test ($p < 0.05$) was chosen due to the non-normal distribution of the genus level data despite attempted transformations. Due to the challenge of zero values in microbial composition [33, 37], a pseudo-count value of one was added to all read counts, then the relative abundance was derived for analysis. The analysis was carried out using R statistical software version 3.6.3. [24].

## Results

### Metabolomic analysis

**Multivariate analysis (MVA).** PCA did not display clear separation among the N, OW, and OB groups. The supervised approach using PLS-DA also failed to show clear separation. Cross-validation Q2 values were negative regardless of the number of principal components, strongly suggesting that the model lacked predictive power or that it was overfitted. This was attributed to noise in the data and a relative small sample size.

**Univariate analysis (UVA).** Sixty-three distinct metabolites were identified through com-pound deconvolution (Fig 1). One-way ANOVA identified five significantly different com-pounds among the three groups: butyrate ($p = 0.016$), arabinose ($p = 0.033$), galactose ($p = 0.036$), trimethylamine (TMA) ($p = 0.044$), and acetate ($p = 0.045$). After application of Tukey HSD post hoc test all compounds, except for acetate ($p = 0.063$), showed a significant difference between the N and OB groups and none showed significance between N and OW and between OW and OB groups. All of these compounds had a higher mean concentration in the OB group compared to the N group (Fig 2). Application of the false discovery rate (FDR) for multiple comparisons suggested that only 44% of these five metabolites would be expected to be significant. Given the study's sample size and not to discard potentially valuable

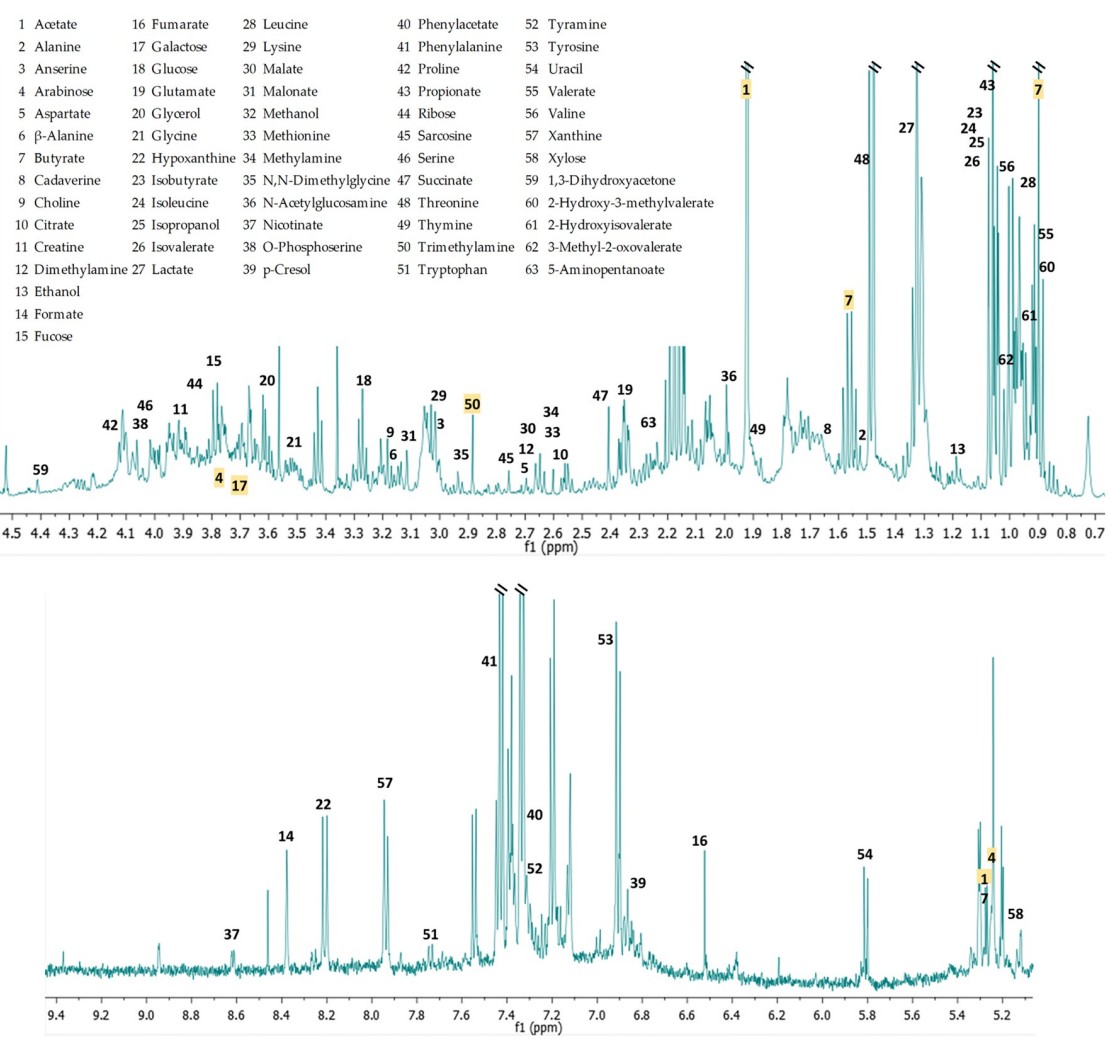

**Fig 1. Representative ¹H NMR spectrum.** 63 identified metabolites. Compounds in yellow are the five significantly different (p < 0.05) metabolites based on one-way ANOVA. For visual clarity, with the exception of the five significant metabolites, compounds are only listed once in the spectra regardless of their actual number of spectral peaks. The water region (4.5 to 5.2 ppm) has been excluded.

metabolites that may be important for generating further hypotheses, we have included the metabolites identified through the Tukey HSD test in the Discussion section.

A Pearson correlation of these metabolites between themselves and the z-score showed the following as significant: z-score with arabinose (p = 0.050, correlation coefficient (cc) = 0.31), galactose (p = 0.014, cc = 0.38), and TMA (p = 0.016, cc = 0.34); acetate with butyrate (p < 0.001, cc = 0.73) and TMA (p = 0.004, cc = 0.40); and arabinose with galactose (p < 0.001, cc = 0.67). These all displayed a positive relationship with the strongest correlations between acetate with butyrate, and arabinose with galactose.

## 16S rDNA analysis

A total of 83 genus, 36 family, 19 order, 15 class, and 6 phylum level taxa were identified. Fig 3 is a heatmap based on the phyla's relative abundance among the participants in the three study groups. Alpha-diversity assessment at the genus, family, and phylum levels showed no

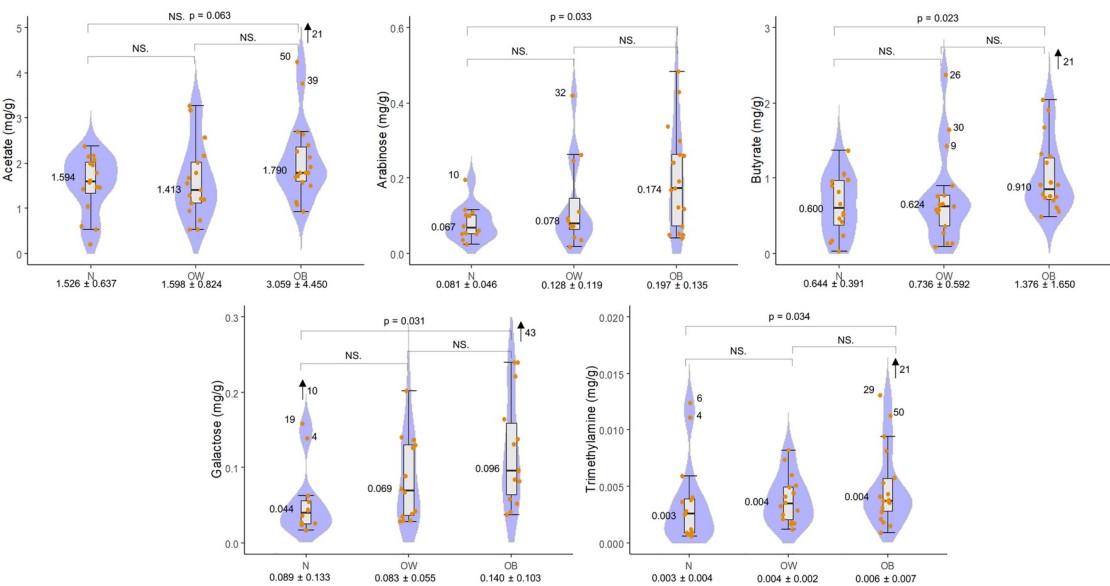

**Fig 2. Metabolite concentration (mg/g) boxplots.** Significantly different (p < 0.05) metabolites among the N, OW, and OB groups based on one-way ANOVA (significance found only between N and OB groups after post-hoc test. Acetate was not significant after application of post-hoc test). The x-axis shows the group name and the mean ± standard deviation. The numbers and text in the graphical area represent: the post-hoc p-value where significant, NS. = not significant, the median and the sample numbers that lie outside the visible range area.

significant differences (all p-values > 0.3) among the N, OW, and OB groups. Likewise, the Firmicutes to Bacteroidetes (F/B) ratio was not significantly different among the three groups by one-way ANOVA testing.

Differential abundance analysis through ANCOM did not identify any significant taxa at any of the taxonomic levels, whereas the more permissive EdgeR identified the following suggestive associations: *Escherichia* (p = 0.005), and *Tyzzerella* subgroup 3 (p = 0.006) at the genus level; the signal from *Escherichia* was reflected also at the family level of *Enterobacteriaceae* (p = 0.009). No differential abundance was noted at the order, class, nor phylum levels. *Escherichia* is one of the main representatives of *Enterobateriaceae* [38], and both taxa displayed a similar relative abundance pattern by showing a decrease from the N to the OB group; in contrast, *Tyzzerella* subgroup 3 showed an increase. Fig 4 shows the genera with the highest

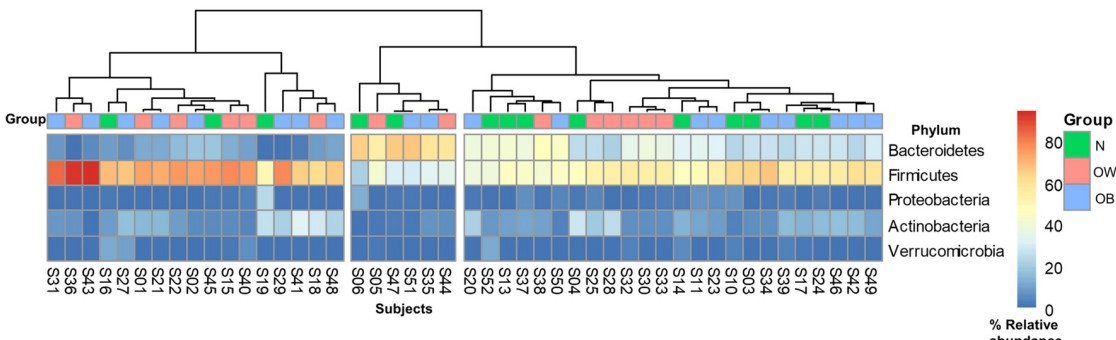

**Fig 3. Heatmap based on phyla % relative abundance.** Warmer colors indicate higher % relative abundance, which was exhibited by Bacteroidetes and Firmicutes. Cooler colors indicate lower % relative abundance. Inter-individual variability does not display clear clustering among the three study groups.

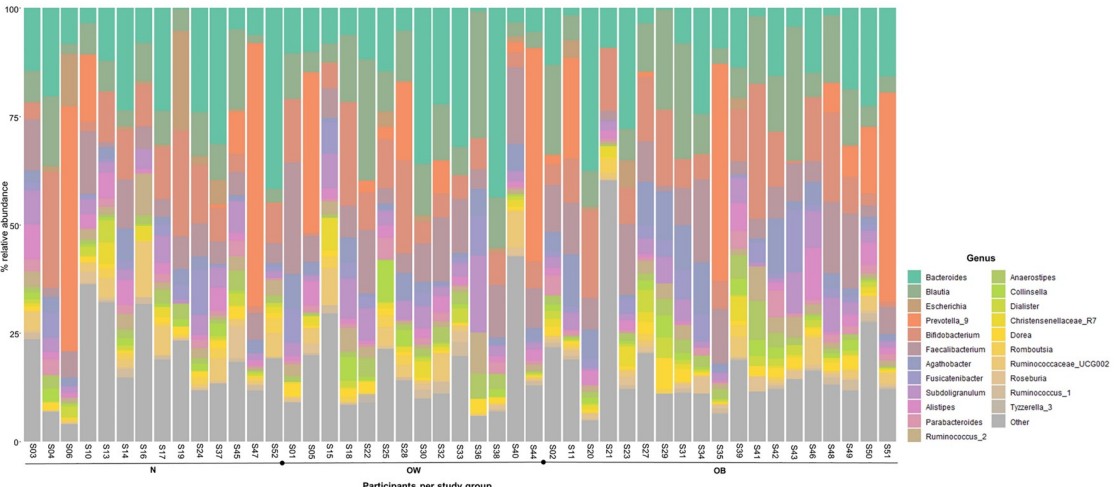

**Fig 4. Stacked bar charts based on genera % relative abundance.** Display of the 20 most abundant genera (by relative abundance), as well as the two significant genera (*Escherichia* and *Tyzzerella* subgroup 3) by study group. Genus Other represents the aggregate of the remaining 61 identified genera.

relative abundances, as well as *Escherichia* and *Tyzzerella* subgroup 3. Fig 5 displays the log10 transformed relative abundance for the two significant genera in the three study groups.

## Correlation of metabolomic and 16S rDNA analyses

Spearman correlation of the relative abundance of the significant taxa with the concentration of the significant metabolites and with the z-score did not show any positive significant correlations. Nevertheless, as shown in Table 2, strong positive and negative correlations were observed among other significant metabolites and non-significant genera, as well as the reverse. Fig 6 displays the strength of the positive and negative correlations among all identified genera and metabolites.

## Discussion

Our results showed an increase of fecal butyrate in the OB compared to the N group, which lends support to previous observations of higher short-chain fatty acid (SCFA) concentrations

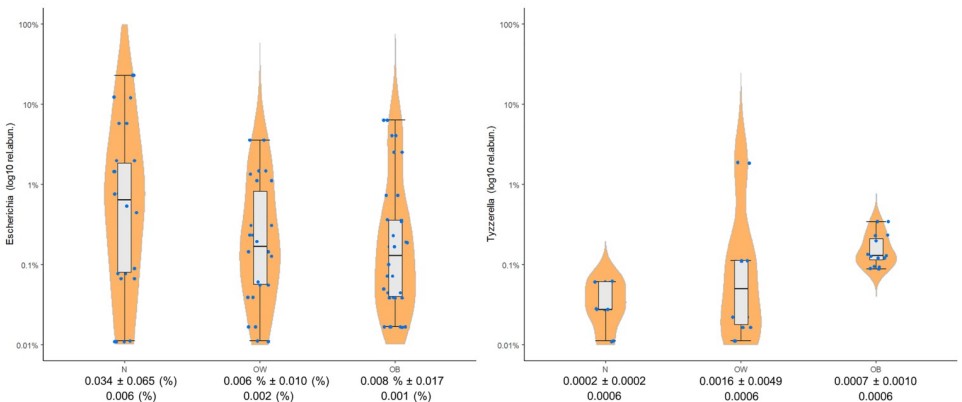

**Fig 5. Genera relative abundance boxplots.** The two genera that displayed significant difference among the N, OW, and OB groups. The x-axis shows the group name and the mean ± standard deviation, and underneath is the median.

**Table 2. Significant correlations of the significant metabolites with identified genera.**

| Genus | Butyrate | Arabinose | Galactose | TMA |
|---|---|---|---|---|
| *Blautia* | | | 0.425 | |
| *Butyricicoccus* | | | 0.326 | |
| *Butyricimonas* | | - 0.346 | | |
| *Catenibacterium* | | | | - 0.364 |
| *Coprococcus* 1 | - 0.481 | | | |
| *Coprococcus* 3 | | | | - 0.303 |
| *Desulfovibrio* | - 0.413 | | | - 0.428 |
| *Eggerthella* | - 0.290 | | | |
| *Erysipelotrichaceae* UCG-003 | | 0.457 | 0.450 | |
| *Fusinibacter* | | | 0.409 | |
| *Haemophilus* | 0.420 | | | |
| *Paraprevotella* | - 0.300 | | | - 0.311 |
| *Parasutterella* | | | | - 0.301 |
| *Romboutsia* | | | - 0.339 | |
| *Roseburia* | 0.304 | | | |
| *Ruminoclostridium* 5 | - 0.477 | | | - 0.398 |
| *Ruminoclostridium* 6 | - 0.317 | | - 0.327 | |
| *Ruminoclostridium* 9 | | | - 0.430 | - 0.377 |
| *Ruminococcaceae* NK4A214 | | | - 0.472 | - 0.360 |
| *Ruminococcaceae* UCG-002 | | | - 0.366 | |
| *Ruminococcaceae* UCG-003 | | | - 0.394 | |
| *Ruminococcaceae* UCG-010 | | | - 0.355 | - 0.354 |
| *Slackia* | - 0.349 | | | |

Values are Spearman correlation coefficients. TMA = trimethylamine.

in children with overweight/obese compared to those who are normal-weight [8, 39–41]. An increased SCFA concentration, especially butyrate and acetate, has also been observed in obese mice when compared to their lean counterparts [42]. Two suggested reasons for this are: 1) higher substrate fermentation activity by gut microbiota, which translates into increased microbial energy harvest, and/or 2) decreased absorption due to either low-grade inflammation, more rapid gut transit time, or shifts in microbial cross-feeding patterns [8, 41, 43]. Given that the diet among the participants in our study was, except for portion size (OW and OB consumed, on average, 30% fewer kcal than N), approximately homogenous (approx. 17% protein, 28% fat and 55% carbohydrates), it does suggest higher microbial fermentation activity from fermentative substrates such as resistant starch and dietary fiber, the main sources of microbiota-derived SCFAs [5, 8, 44]. Despite consuming less kcal, the OW and OB groups showed significantly more fecal butyrate, which has been identified as the main energy supplier for colonic epithelial cells [8]. It is common for microbiota produced butyrate to end up in stool when not consumed by the colonic epithelium [8]. It has been estimated that SCFAs contribute about 60–70% of the energy requirements of colonic epithelial cells and 5–15% of the total caloric requirements of humans [45]. A proposed mechanism on how an increase in butyrate and other SCFAs may increase energy harvest is that SCFAs may serve as substrates for hepatic de novo lipogenesis (DNL) [42, 46]. The excess non-metabolized SCFAs reach the liver via the portal system, where they may serve as precursors for gluconeogenesis in case of propionate, and lipogenesis for acetate and butyrate [47, 48]. Goffredo et al. (2016) in a study of 84 youth ranging from non-obese to severely obese observed that the three major SCFAs

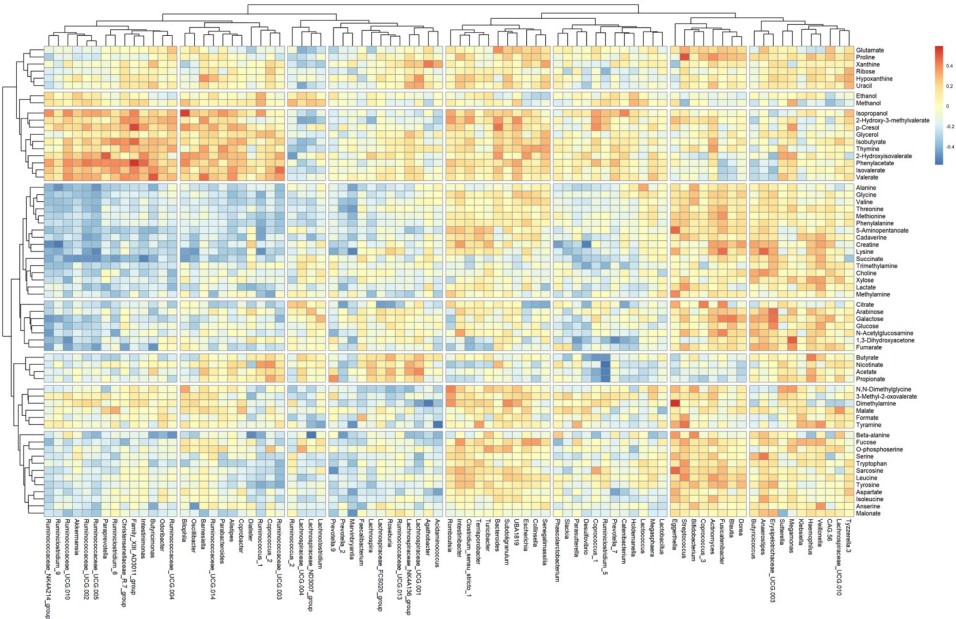

**Fig 6. Metabolite-genus spearman correlation heatmap.** The x-axis shows the genera and the y-axis the metabolites. Warmer colors indicate positive correlations. Cooler colors indicate negative correlations. The more intense the color, the closer the number is to the Spearman correlation value 1 or -1.

were positively associated with body and visceral fat, and from these, butyrate was the only one significantly associated with hepatic fat; furthermore, when a subset of this group was tested for associations with DNL, butyrate was significantly associated with a delta increase in hepatic DNL after a controlled dietary carbohydrate load [46]. The same study, using an *in vitro* stool assay, also observed a higher fermentation capability of fructose by youth with obesity compared to nonobese, which further supports the concept of increased energy harvest from food in those youth with obesity [42, 46]. In looking for the potential instigators of these changes, the gut microbiota, it is important to keep in perspective that the SCFA-producing microbes are a phylogenetically diverse group [45], with a wide distribution of enteric bacteria producing acetate; a much more conservative distribution for butyrate, the most well-known being in the Firmicutes phylum (*Faecalibacterium*, *Eubacterium*, *Roseburia*); and for propionate several Firmicutes, Bacteroidetes, and Proteobacteria phyla such as families *Veillonellaceae* and *Lachnospiraceae* [49–51]. The two significant taxa in our study, *Escherichia* and *Tyzzerella* subgroup 3, were not significantly associated with butyrate; nevertheless, the SCFA was significantly associated with nine genera (Table 2). Butyrate's two positive correlations, with *Haemophilus* and *Roseburia*, are supported by several previous studies [7, 49, 50, 52].

Although the higher concentration of SCFAs in children with overweight/obese in several studies [8, 39–41], including butyrate in ours, could suggest them as an obesity biomarker, this is not without controversy given that SCFAs are attributed a myriad of health benefits such as, among others, improvement in blood lipid profiles, glucose homeostasis, and even reduced body weight [7, 53]. How does one reconcile this contradiction? A potential conceptual analogy is that of nutrient overload, a certain nutrient amount may confer benefits, but an excess of it could very well be detrimental. Also, it is important to keep in mind the limitations of each study; for example, the anti-inflammatory effects of butyrate have been studied mainly *in vitro* [45]. In addition, it may be more beneficial to look for biomarkers as part of a panel of biomarkers instead of individual ones as mentioned by Vignoli et al. (2019) [54].

In addition to SCFAs, monosaccharides arabinose and galactose also had higher concentrations in our OW and OB groups, and the two monosaccharides displayed a strong positive correlation with each other, and both showed a significant positive correlation with the BMI z-score. It appears that most monosaccharides in stool often originate from the non-absorbed breakdown of polysaccharides (resistant starches, dietary fiber), which are the main source of carbon and energy for the gut microbiota [55], or directly from the diet which can be used as nutrition by the host's enzymes [56]. A higher concentration of arabinose and galactose in the OW and OB groups may suggest an excess of saccharides from the diet or from the breakdown of polysaccharides which are not absorbed due to energy needs being met without them. Given the controlled diet of the participants, the results thus suggest the origin to be the polysaccharide breakdown. This would directly tie in with the concept of an increased energy harvest by the microbial dysbiosis in the obese state [8, 41, 43], which would result in an excess of monosaccharides as well as a higher load of SCFAs.

Trimethylamine (TMA) also showed higher concentrations in the OW and OB groups. It is known to be produced by various gut microbiota taxa from dietary quaternary amines, mainly choline and L-carnitine derived from eggs, milk, liver, red meat, poultry, shell fish and fish [57–59]. It's considered toxic due to its further oxidation into trimethylamine N-oxide (TMAO), which has been associated with atherosclerosis, cardiovascular diseases, and other ailments [57, 59, 60]. In a previous study, children with obesity showed a decrease of TMA in fecal water after a diet intervention consisting of rich amounts of non-digestible carbohydrates [60]. TMA was also shown to be downregulated in the urine of children supplemented with non-digestible carbohydrates [6, 61]. Other diet induced changes included significant weight loss, structural microbiota changes, a reduction of serum antigen load, and alleviation of inflammation [6]. The identified taxa involved in TMA production appear to constitute members of the core gut community, though at very low abundances and characterized by functional redundancies indicating that several taxa potentially contribute to the TMA pool [60]. The majority of these were members of the genus *Clostridium* XIVa and a specific *Eubacterium* [60]. This potential is further supported by TMA's significant positive correlation with acetate (p = 0.004) and the BMI z-score (p = 0.016).

Genera *Escherichia* (phylum Proteobacteria), and *Tyzzerella* subgroup 3 (phylum Firmicutes), were not significantly correlated with our significant metabolites, but they were significantly different among the three study groups. In our study, *Escherichia* decreased in relative abundance from the N to the OB group. The genus includes both commensal and pathogenic species, and although species *E.coli* has been observed to be increased in children with obesity compared those with normal-weight [62], a general pattern of this genus in relation to childhood obesity is still very open to investigation. With *Tyzzerella* subgroup 3 we observed a relative abundance increase from the N to the OB group. We could not find any associations between this taxon and obesity in the literature and, overall, this member of *Lachnospiraceae* appears to be of little medical relevance; nevertheless, it has been reported in connection to dietary variables [63], and one study did observe related genera *Tyzzerella* and *Tyzzerella* subgroup 4 to be enriched in a group of adults with higher cardiovascular disease risk when compared to lower-risk subjects [64]. Even though our 16S rDNA analysis only revealed two significantly different genera among the three study groups, it is important to point out, especially for future investigations, that the obese phenotype may be better characterized by the abundance of several distinct communities rather than by the presence of specific species [46]; furthermore, an alteration in efficiency of energy harvest produced by gut bacterial composition changes does not have to be great to contribute to obesity given that small changes in energy balance, over the course of a year, can result in significant changes in body weight [42, 65]. In addition, another possibility is that although inter-group compositional differences

may be minimal, the differences observed in our metabolite data could rather indicate differences in bacterial functional activity where metabolically versatile species adapt to changing nutritional circumstances by selectively metabolizing some substrates to the exclusion of others, thus affecting the types and amounts of fermentation products produced from substrates [55]. An apparent change in microbiota functionality, but not in composition, was observed in a study my Morales et al. (2016) where a high-fat diet accompanied by fiber supplementation induced inflammation while not altering gut microbiota composition [66].

In conclusion, our findings suggest support to the hypothesis of increased energy harvest in obesity by the human gut microbiota through the growing observation of increased fecal butyrate in children with overweight/obesity and an increase of certain monosaccharides in the stool. Also supported is butyrate's positive correlation with *Haemophilus* and *Roseburia*, as well as the increase of trimethylamine as an indicator of an unhealthy state.

## Supporting information

**S1 Table. Characteristics of the 52 participants.** T1 and T2 refer to one and two days prior to the sampling date. The kcal amount is per the entire day.
(XLSX)

## Acknowledgments

We warmly thank the study participants and their families, as well as the staff from Olivova Children's Medical Institution (Olivova Dětská Léčebna), whose support was essential in completing this investigation.

## Author Contributions

**Conceptualization:** Andrea Slavíčková, Karel Černý, Jaroslav Havlík.

**Data curation:** Ondřej Cinek, Jaroslav Havlík.

**Formal analysis:** José Diógenes Jaimes, Andrea Slavíčková, Jaroslav Havlík.

**Funding acquisition:** Jaroslav Havlík.

**Investigation:** José Diógenes Jaimes, Andrea Slavíčková, Lucie Vodolánová, Jaroslav Havlík.

**Methodology:** José Diógenes Jaimes, Andrea Slavíčková, Jakub Hurych, Ondřej Cinek, Jaroslav Havlík.

**Project administration:** Jaroslav Havlík.

**Resources:** Ondřej Cinek, Jaroslav Havlík.

**Software:** José Diógenes Jaimes, Jaroslav Havlík.

**Supervision:** Jaroslav Havlík.

**Validation:** Ben Nichols.

**Visualization:** José Diógenes Jaimes.

**Writing – original draft:** José Diógenes Jaimes.

**Writing – review & editing:** Ondřej Cinek, Jaroslav Havlík.

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
