## [Decision Letter · Decision Letter 0]

26 Oct 2020

PONE-D-20-27287

Stool metabolome-microbiota evaluation among children and adolescents with obesity, overweight, and normal-weight using 1H NMR and 16S rRNA gene profiling

PLOS ONE

Dear Dr. Jaimes,

Thank you for submitting your manuscript to PLOS ONE. After careful consideration, we feel that it has merit but does not fully meet PLOS ONE’s publication criteria as it currently stands. Therefore, we invite you to submit a revised version of the manuscript that addresses the points raised during the review process.

Both reviewers agreed that more or somewhat different analysis could benefit the manuscript.  I agree with reviewer 1's concerns about the specific components of the different diets, and the nuance that is creating in microbiomes by nutrient type, bioavailability, and preparation effects.  I encourage the authors to provide significantly more detail on the diets to address this point.

Reviewer 2 suggested additional citations, which were accidentally omitted in their submitted review.  These include:

Shankar, V., Homer, D., Rigsbee, L. et al. The networks of human gut microbe–metabolite associations are different between health and irritable bowel syndrome. ISME J 9, 1899–1903 (2015). https://doi.org/10.1038/ismej.2014.258

and

   Cribbs, S.K., Uppal, K., Li, S. et al. Correlation of the lung microbiota with metabolic profiles in bronchoalveolar lavage fluid in HIV infection. Microbiome 4, 3 (2016). https://doi.org/10.1186/s40168-016-0147-4

We look forward to receiving your revised manuscript.

Kind regards,

Suzanne L. Ishaq, PhD

Academic Editor

PLOS ONE

Journal Requirements:

3. In your Methods section, please provide additional information about the participant recruitment method and the demographic details of your participants. Please ensure you have provided sufficient details to replicate the analyses such as: a) the recruitment date range (month and year), b) a description of any inclusion/exclusion criteria that were applied to participant recruitment, c) a table of relevant demographic details, c) a description of how participants were recruited.

4.  We suggest you thoroughly copyedit your manuscript for language usage, spelling, and grammar. If you do not know anyone who can help you do this, you may wish to consider employing a professional scientific editing service.  

5. Please provide a sample size and power calculation in the Methods, or discuss the reasons for not performing one before study initiation.

Reviewers' comments:

Reviewer's Responses to Questions

**Comments to the Author**

1. Is the manuscript technically sound, and do the data support the conclusions?

Reviewer #1: Partly

Reviewer #2: Yes

2. Has the statistical analysis been performed appropriately and rigorously? 

Reviewer #1: Yes

Reviewer #2: Yes

3. Have the authors made all data underlying the findings in their manuscript fully available?

Reviewer #1: No

Reviewer #2: Yes

4. Is the manuscript presented in an intelligible fashion and written in standard English?

Reviewer #1: Yes

Reviewer #2: Yes

5. Review Comments to the Author

Reviewer #1: Summary – This study examines associations between microbial taxa, based on 16S rRNA sequencing, and the stool metabolome, via 1H NMR, in normal, overweight, and obese individuals. While interesting, the associations of both microbial taxonomy and metabolites with obesity is well described in the literature. This manuscript offers no new concepts, and significant issues must be addressed.

Conceptual/Major Comments –

1. Given the focus on obesity, metabolites, and diet, the description of diet in this study is lacking on multiple levels.

a. The statement that diet was controlled in this study by simply balancing macronutrients (fats, carbohydrates, and protein) is a misleading overstatement. Given these constraints, one subject could have eaten simple carbohydrates (e.g., highly processed white bread) and another subject could have eaten an equivalent caloric percentage of carbohydrates of completely complex carbohydrates (e.g., fruits, vegetables), and these would be considered the same; however, these diets would lead to extremely different microbiota and metabolomes.

b. Additionally, there is no description of calorie controls beyond the statement that “the children with obese/overweight on a caloric restriction”. Given that half of the metabolites identified (arabinose, galactose) are found in food and do not require transformation by microbial metabolism, a highly plausible explanation is that subjects in the OW and OB groups consumed more food that resulted in the higher amounts of arabinose and galactose in the stool. If diet is to be considered in this manuscript, there must be a complete explanation and analysis of diet as a co-factor.

2. The control group (normal weight) is a group that was receiving treatment of respiratory or locomotive conditions. Individuals with underlying and potentially confounding conditions should not be used as controls.

3. A complete description, or reference, is needed to describe the mock community used as a reference.

4. The relationship between gut microbiome and obesity has been examined extensively in many studies with large sample sizes. For a summary see:

Sze, M.A. and Schloss, P.D., 2016. Looking for a signal in the noise: revisiting obesity and the microbiome. MBio, 7(4), pp.e01018-16. Additional validation of these trends is not needed.

The same holds for the SCFA connection, for example

Schwiertz, A., Taras, D., Schäfer, K., Beijer, S., Bos, N.A., Donus, C. and Hardt, P.D., 2010. Microbiota and SCFA in lean and overweight healthy subjects. Obesity, 18(1), pp.190-195.

Kim, K. N., Yao, Y., & Ju, S. Y. (2019). Short Chain Fatty Acids and Fecal Microbiota Abundance in Humans with Obesity: A Systematic Review and Meta-Analysis. Nutrients, 11(10), 2512. https://doi.org/10.3390/nu11102512

The authors should consider refocusing the analyses on the age of the subjects (most studies have been conducted in adults) and/or additional types of analyses that have not been previously conducted.

5. Speculations regarding mechanisms (i.e., increased energy harvest) are lacking support and should be revised.

6. Please consider changing the type of figure used to display taxonomic abundance (Figure 4). The pie charts currently used are an ineffective and inaccurate method of representing the data, as they do not include any display of variance. Consider using a heatmap to show intersubject variability. Additionally, the Firmicutes/Bacteroidetes appear to be incorrect based on the data presented in this figure (N – 50.4/34.1 = 1.48, not 2.2; OW – 57.9/28.7 = 2.02, not 3.4; OB – 59.3/26.2 = 2.26, not 5.7).

7. Tukey’s HSD provides weak control of Type I error. Please consider using false discovery rate to correct for multiple comparisons in this type of exploratory study.

Reviewer #2: In this article the authors aim to show a difference in stool microbiome bacterial composition and metabolomic profile between children with normal, overweight, and obese BMI z-scores. They also attempted to relate the bacterial relative abundance with the metabolite abundance. Among the bacterial composition, the authors find 2 “suggestive associations” (I really like this wording): decreased Escherichia in the obese group compared to the normal group, and increased Tyzzerella subgroup 3 in the obese group compared to the normal group. Among the metabolites, the authors find 5 that are significantly higher in the obese group than the normal group: butyrate, arabinose, galactose, trimethylamine, and acetate. These increased metabolites support for the hypothesis that the gut microbiome in obese people has increased energy harvest compared to people of normal weight.

Overall, the authors have a well designed and appropriately analyzed study that further supports the increased energy hypothesis that others have put forth. I found the discussion of the controversy around higher SCFAs as a biomarker (lines 333-342) particularly clear and helpful. I have a couple of suggestions that would improve the manuscript and some minor concerns that I would like to see addressed before publication.

Major Suggestions:

I would like to see a more sophisticated analysis to find associations between microbes and metabolites. The authors only examined microbes and metabolites that were significantly different across BMI groups for positive Spearman correlations (page 12, line 313). An analysis that includes all metabolites and bacteria could reveal other patterns that differ across BMI groups; examples include correlations (seen in Shankar, et al) and sparse partial least squares regression (seen in Cribbs, et al).

You refer to the microbiome but only examined bacteria. A discussion that includes other microbes, or the caveat that only bacteria were examined, would widen the audience and appeal of the study.

Minor Concerns:

Lines 77-81: The sentence that starts “For example” is difficult to read and should be simplified to avoid having so many lists in one sentence.

The correlations between metabolites (lines 175-178, 276-284, and fig 3) doesn’t seem to add anything and I would suggest removing it.

Line 211 - When you removed cyanobacteria, were these all cyanobacteria or only those that could not be classified further (potentially indicating non-bacterial origins)? If the latter, please specify.

Line 237 - There’s an extra comma between “or greater” and “a variance >10%”.

Line 340 - This is the second sentence in a row to start with “Also”. Consider using “Finally”, another transition word, or nothing at all.

Line 345 - I would suggest you move the reference to Fig 3 to after “with each other” which is what is displayed in the figure - if you decide to keep it, see above.

Line 370 - Please specify that it is BMI z-score.

6. PLOS authors have the option to publish the peer review history of their article (what does this mean?). If published, this will include your full peer review and any attached files.

Reviewer #1: **Yes: **Elliot S Friedman

Reviewer #2: **Yes: **Laura Tipton

---

## [Author Response · Author response to Decision Letter 0]

27 Nov 2020

Response to reviewers.

Reviewer 1:

Comment:

1. Given the focus on obesity, metabolites, and diet, the description of diet in this study is lacking on multiple levels.

a. The statement that diet was controlled in this study by simply balancing macronutrients (fats, carbohydrates, and protein) is a misleading overstatement. Given these constraints, one subject could have eaten simple carbohydrates (e.g., highly processed white bread) and another subject could have eaten an equivalent caloric percentage of carbohydrates of completely complex carbohydrates (e.g., fruits, vegetables), and these would be considered the same; however, these diets would lead to extremely different microbiota and metabolomes. 

Response: 

We agree with your comment and have now provided more detailed information about the diet that the participants followed. The composition of the meals among the participants was similar (similar ingredients, similar dishes); for example, the carbohydrate portion of each meal was of the same food item, differing only in the portion size based on the subject’s age, gender, and weight. This information has now been more explicitly stated in the methods subsection “Study participants.” Additionally, new information includes the macronutrient percentage intake per each of the three groups (normal, overweight, obese) one and two days prior to sampling (Table 1), as well as per each of the participants (S1 Table 1). This has now been more clearly stated (lines 127-133). 

Comment:

b. Additionally, there is no description of calorie controls beyond the statement that “the children with obese/overweight on a caloric restriction”. Given that half of the metabolites identified (arabinose, galactose) are found in food and do not require transformation by microbial metabolism, a highly plausible explanation is that subjects in the OW and OB groups consumed more food that resulted in the higher amounts of arabinose and galactose in the stool. If diet is to be considered in this manuscript, there must be a complete explanation and analysis of diet as a co-factor.

Response: 

We have now explicitly stated that those with overweight/obesity were in a 30 % caloric restriction (lines 129-130). Additionally, daily kilocalorie intake per each of the three groups (normal, overweight, obese) one and two days prior to sampling (Table 1), as well as per each of the participants (S1 table 1), has now been included. The diet has been more clearly presented through these tables and in the methods section (lines 127-133). We believe that the new provided information more accurately reflects the consideration of diet as a co-factor and the potential bias it may introduce in the interpretation. 

Comment:

The control group (normal weight) is a group that was receiving treatment of respiratory or locomotive conditions. Individuals with underlying and potentially confounding conditions should not be used as controls.

Response:

We agree that the control group (normal-weight youth) may not be an ideal control group; however, given the selection criteria, we still believe that they represent a valuable comparison group. The controls with a respiratory disease were in remission of their condition, and those with an orthopedic (locomotive) diagnoses were healthy enough to take part in the program’s physical activity regime. In addition, the recruitment criterion was no antibiotics in the past three months prior to program participation, none were taking medications, and all were physically able to take part in a physical activity program. Furthermore, except for portion size, they all received a similar diet (same ingredients, same dishes). As a result of the selection criteria, a similar physical activity regime, a homogenous diet among the participants, and the same ethnic and geographic background of the participants, the effect of cofounding variables was minimized. Consequently, we believe that the observations from our study are relevant in contrasting the metabolomic and gut microbiota differences among the three groups (normal-weight, overweight, obese) in children/adolescents. The section describing this (lines 113-135) has been updated to provide these and further details. 

Comment:

A complete description, or reference, is needed to describe the mock community used as a reference.

Response:

Further details have been provided to describe all the participants, including the controls. (lines 113-135, Table 1, S1 Table 1)

Comment:

The relationship between gut microbiome and obesity has been examined extensively in many studies with large sample sizes. For a summary see:

Sze, M.A. and Schloss, P.D., 2016. Looking for a signal in the noise: revisiting obesity and the microbiome. MBio, 7(4), pp.e01018-16. Additional validation of these trends is not needed.

The same holds for the SCFA connection, for example

Schwiertz, A., Taras, D., Schäfer, K., Beijer, S., Bos, N.A., Donus, C. and Hardt, P.D., 2010. Microbiota and SCFA in lean and overweight healthy subjects. Obesity, 18(1), pp.190-195.

Kim, K. N., Yao, Y., & Ju, S. Y. (2019). Short Chain Fatty Acids and Fecal Microbiota Abundance in Humans with Obesity: A Systematic Review and Meta-Analysis. Nutrients, 11(10), 2512. https://doi.org/10.3390/nu11102512

The authors should consider refocusing the analyses on the age of the subjects (most studies have been conducted in adults) and/or additional types of analyses that have not been previously conducted.

Response:

Although it has been examined, the findings are still far from being established. There is ample room for expansion and provide a more complete and precise picture of the metabolomic-gut bacteria changes involved. 

In regard to focusing the analyses on the age of the subjects, that is actually one of our aims since, as you mentioned, most students have been focused on adults. To express this aim more explicitly, we have revised the text (lines 65, 75-77, 87-91). This included the addition of an additional citation (Radjabzadeh D, Boer CG, Beth SA, van der Wal P, Kiefte-De Jong JC, Jansen MAE, et al. Diversity, compositional and functional differences between gut microbiota of children and adults. Sci Rep. 2020;10: 1–13. doi:10.1038/s41598-020-57734-z).

Comment:

Speculations regarding mechanisms (i.e., increased energy harvest) are lacking support and should be revised.

Response:

We have elaborated the Discussion section to provide more support for the increased energy harvest mechanism in lines 374-376 as well as other details throughout this section. We do attempt to make it clear in our writing that this is still a proposed mechanism. Several citations (some previously in the manuscript and some new additions) support this mechanism. (Turnbaugh P. J., Ley R. E., Mahowald M. A., Magrini V., Mardis E. R. GJI. An obesity-associated gut microbiome with increased capacity for energy harvest. Nature. 2006;444: 1027–1022. doi:10.1007/s11837-013-0766-1), (Goffredo M, Mass K, Parks EJ, Wagner DA, McClure EA, Graf J, et al. Role of gut microbiota and short chain fatty acids in modulating energy harvest and fat partitioning in youth. J Clin Endocrinol Metab. 2016;101: 4367–4376. doi:10.1210/jc.2016-1797), (Oliphant K, Allen-Vercoe E. Macronutrient metabolism by the human gut microbiome: Major fermentation by-products and their impact on host health. Microbiome. 2019;7: 1–15. doi:10.1186/s40168-019-0704-8), (Morrison DJ, Preston T. Formation of short chain fatty acids by the gut microbiota and their impact on human metabolism. Gut Microbes. 2016;7: 189–200. doi:10.1080/19490976.2015.1134082), (Riva A, Borgo F, Lassandro C, Verduci E, Morace G, Borghi E, et al. Pediatric obesity is associated with an altered gut microbiota and discordant shifts in Firmicutes populations. Environ Microbiol. 2017;19: 95–105. doi:10.1111/1462-2920.13463), (Schwiertz A, Taras D, Schäfer K, Beijer S, Bos NA, Donus C, et al. Microbiota and SCFA in Lean and Overweight Healthy Subjects. Obesity. 2010;18: 190–195. doi:10.1038/oby.2009.167), (Payne AN, Chassard C, Zimmermann M, Müller P, Stinca S, Lacroix C. The metabolic activity of gut microbiota in obese children is increased compared with normal-weight children and exhibits more exhaustive substrate utilization. Nutr Diabetes. 2011;1: e12–e12. doi:10.1038/nutd.2011.8).

Comment:

Please consider changing the type of figure used to display taxonomic abundance (Figure 4). The pie charts currently used are an ineffective and inaccurate method of representing the data, as they do not include any display of variance. Consider using a heatmap to show intersubject variability. Additionally, the Firmicutes/Bacteroidetes appear to be incorrect based on the data presented in this figure (N – 50.4/34.1 = 1.48, not 2.2; OW – 57.9/28.7 = 2.02, not 3.4; OB – 59.3/26.2 = 2.26, not 5.7).

Response:

We followed your suggestion and have replaced this figure with that of a heat map to better display inter-individual variability. Since there was no significant difference among the three groups, the percentages from the previous figure were excluded in this figure. The previous Firmicutes/Bacteroidetes discrepancy was due to a calculation that included the outliers that had been excluded in the previous figure. 

Comment:

Tukey’s HSD provides weak control of Type I error. Please consider using false discovery rate to correct for multiple comparisons in this type of exploratory study.

Response:

We have considered this and did notice that our significant metabolites were significant at a false discovery rate of 0.56, which means that we would expect 44 % of the identified metabolites to be significant, except that it is hard to ascertain which ones. Since it is an exploratory study, we would like to keep our current methodology as to not potentially discard valuable metabolites. Other researchers may find this important for generating further hypotheses. 

Reviewer 2:

Comment:

I would like to see a more sophisticated analysis to find associations between microbes and metabolites. The authors only examined microbes and metabolites that were significantly different across BMI groups for positive Spearman correlations (page 12, line 313). An analysis that includes all metabolites and bacteria could reveal other patterns that differ across BMI groups; examples include correlations (seen in Shankar, et al). The sparse partial least squares regression (seen in Cribbs, et al) 

Response:

We believe that you raise a very important point, and we also agree that an analysis that includes all metabolites and genera would be very valuable to report. We consulted both of the examples you provided, and we proceeded with a Spearman correlation analysis similar to what we had initially done; however, we now included all metabolites and genera detected and have displayed these correlations in the form of a heatmap (Fig 5) in the results subsection “Correlation of metabolomic and 16S rDNA analyses.” Consequently, we have also added some material both in the results and discussion sections.

Comment:

You refer to the microbiome but only examined bacteria. A discussion that includes other microbes, or the caveat that only bacteria were examined, would widen the audience and appeal of the study.

Response: 

We have edited our language throughout the paper to reflect that only bacteria were examined. We still kept the same title, but throughout the text we believe that it is now clearer that we specifically examined bacteria. We agree that the appeal would be widened by including microbiota outside of bacteria; however, our 16s analysis results are better suited to report and focus only on bacteria for this study. 

Comment:

Lines 77-81: The sentence that starts “For example” is difficult to read and should be simplified to avoid having so many lists in one sentence.

Response:

A slight modification of the sentence was made (now line 83-85)

Comment:

The correlations between metabolites (lines 175-178, 276-284, and fig 3) doesn’t seem to add anything and I would suggest removing it.

Response:

We agree with this suggestion and have removed the figure. We are now reporting the correlation coefficients from the figure in written form in lines 311-316. 

Comment:

Line 211 - When you removed cyanobacteria, were these all cyanobacteria or only those that could not be classified further (potentially indicating non-bacterial origins)? If the latter, please specify.

Response:

We removed all identifiable cyanobacteria signal. Its overall share on the total bacteriome was negligible, only 0.077 % reads. It originated from 5 samples, with the largest signal being 2.2% per sample, and second largest being 0.5 % per sample. These cyanobacteria were classifiable only to the level of order, but all belonged to Gastranaerophilales, an order that - rather surprisingly - consistently lacks both photosynthetic, and aerobic pathways [Rochelle M Soo et al, Science 2017, vol 355, issue 6332, pp. 1436-1440]. The lack of aerobic metabolism makes it probable that the finding is not incidental: these cyanobacteria may indeed be present in the gastrointestinal tract, as the order name suggests and as has been documented by Soo et al in an earlier submission to the Sequencing Reads Archives (https://www.ncbi.nlm.nih.gov/bioproject/PRJNA348149). However, we think it is prudent to delete this negligible portion of reads: (a) the power to detect an association with the outcome is close to zero due to the low abundance of the bacteria, (b) it would confound readers, as cyanobacterium is mostly thought of as a plant predecessor, an organism that can photosynthesize; thus it is often automatically removed like we did.

Comment:

Line 237 - There’s an extra comma between “or greater” and “a variance >10%”.

Response:

It has been addressed (now line 271).

Comment:

Line 340 - This is the second sentence in a row to start with “Also”. Consider using “Finally”, another transition word, or nothing at all.

Response:

The transition word was changed to “In addition” (now line 413).

Comment:

Line 345 - I would suggest you move the reference to Fig 3 to after “with each other” which is what is displayed in the figure - if you decide to keep it, see above.

Response:

We decided to remove the figure per your previous comment about it.

Comment:

Line 370 - Please specify that it is BMI z-score

Response: 

We have now specified that it is BMI z-score (now line 443). Similar specifications were made throughout the manuscript.

Additional requirements:

Comment:

Response:

We have followed the style requirements per the comment, and our revised manuscript reflects this.

Comment:

We note that you have stated that you will provide repository information for your data at acceptance. Should your manuscript be accepted for publication, we will hold it until you provide the relevant accession numbers or DOIs necessary to access your data. If you wish to make changes to your Data Availability statement, please describe these changes in your cover letter and we will update your Data Availability statement to reflect the information you provide.

Response:

The data will be provided via Mendeley Data as we had originally planned under the following DOI link: http://dx.doi.org/10.17632/cwj76cbvc9.1. The items to be included there are: 1) S1 Table 1 (as referenced in the manuscript) containing the individual participant’s gender, age, BMI, BMI z-score, as well as the kilocalorie and macronutrient daily percentage composition one and two days prior to sampling; 2) unrarefied source data for the 16S rRNA gene sequencing analysis; 3) Metabolite concentrations in mg/g derived from Chenomx NMR Suite version 7.5 for each of the 52 study participants; and 4) the 1H NMR spectra for the 52 study participants.

Comment:

In your Methods section, please provide additional information about the participant recruitment method and the demographic details of your participants. Please ensure you have provided sufficient details to replicate the analyses such as: a) the recruitment date range (month and year), b) a description of any inclusion/exclusion criteria that were applied to participant recruitment, c) a table of relevant demographic details, c) a description of how participants were recruited.

Response:

This information has now been more explicitly stated in the methods subsection “Study participants.” 

Comment:

We suggest you thoroughly copyedit your manuscript for language usage, spelling, and grammar. If you do not know anyone who can help you do this, you may wish to consider employing a professional scientific editing service. 

Response:

We have copyedited our manuscript. All revisions can be followed via Track Changes.

Comment:

Please provide a sample size and power calculation in the Methods, or discuss the reasons for not performing one before study initiation.

Response:

Given that this is an exploratory pilot study, we did not have a clear hypothesis about which metrics would be expected to be different among the study groups, thus an accurate power calculation did not seem suitable nor feasible. Also, the study was limited in time and funding, thus we were limited on the cohort size and recruitment time possibilities.

---

## [Decision Letter · Decision Letter 1]

31 Dec 2020

PONE-D-20-27287R1

Stool metabolome-microbiota evaluation among children and adolescents with obesity, overweight, and normal-weight using 1H NMR and 16S rRNA gene profiling

PLOS ONE

Dear Dr. Havlík, ČZU v Praze,

Thank you for submitting your manuscript to PLOS ONE. After careful consideration, we feel that it has merit but does not fully meet PLOS ONE’s publication criteria as it currently stands. Therefore, we invite you to submit a revised version of the manuscript that addresses the points raised during the review process.

The authors have done a great deal of work to revise their manuscript, and the reviewers and I were pleased to see that the changes improved the manuscript.  A few additional considerations have been mentioned which may require some consideration by the authors, and a handful of very minor corrections have been noted.

We look forward to receiving your revised manuscript.

Kind regards,

Suzanne L. Ishaq, PhD

Academic Editor

PLOS ONE

Reviewers' comments:

Reviewer's Responses to Questions

**Comments to the Author**

1. If the authors have adequately addressed your comments raised in a previous round of review and you feel that this manuscript is now acceptable for publication, you may indicate that here to bypass the “Comments to the Author” section, enter your conflict of interest statement in the “Confidential to Editor” section, and submit your "Accept" recommendation.

Reviewer #1: (No Response)

Reviewer #2: All comments have been addressed

2. Is the manuscript technically sound, and do the data support the conclusions?

Reviewer #1: Partly

Reviewer #2: Yes

3. Has the statistical analysis been performed appropriately and rigorously? 

Reviewer #1: No

Reviewer #2: Yes

4. Have the authors made all data underlying the findings in their manuscript fully available?

Reviewer #1: Yes

Reviewer #2: Yes

5. Is the manuscript presented in an intelligible fashion and written in standard English?

Reviewer #1: Yes

Reviewer #2: Yes

6. Review Comments to the Author

Reviewer #1: 1. I appreciate the author’s providing more information about diet. I still have some additional questions/concerns. Was this an outpatient or inpatient study? If inpatient I assume that diet was controlled by the staff (although explicit information would be helpful). However, if this was an outpatient study, how was dietary management implemented? Were meals provided by the study, or were meals prepared by subjects/subject’s families? Additionally, compliance with dietary guidance is a well-established issue in outpatient studies. Was this controlled at all? Finally, how long were subjects on the controlled diet prior to sampling? This would be important to determine whether the observed changes were a response to stimuli (diet) or a new steady-state composition/function of the microbiome and metabolites.

2. I appreciate the additional information regarding control subjects, thank you! It might be interesting to compare reference data sets (e.g., HMP, other pediatric studies) to determine whether the controls in this study are similar to others, but that does not need to be a requirement for publication.

3. I appreciate the additional information on participants, but the comment was in regard to the mock community used for sequencing controls.

5. I appreciate the expanded discussion, thank you! I would recommend revising the wording of the statement “Despite consuming less kcal, the OW and OB groups still produced significantly more butyrate, which has been identified as the main energy supplier for colonic epithelial cells [8].” The measurement in this case is of butyrate in feces – which is butyrate produced by the microbiota but not consumed by the colonic epithelium.

6. This new figure conveys much additional information but is only at the phyla level. It would be helpful to include a genus level figure in addition to (not in replace of) the phyla level results. Additionally, the unsupervised clustering really shows the lack of segregation of microbial communities by group. I would suggest adding to the discussion the notion that while there are minimal differences in microbial community composition between groups, the metabolite data suggests that there may be differences in the metabolic activity of these microbes.

7. Thank you for the response. I think that this is fine but should be mentioned in the manuscript. Perhaps a more appropriate way to address this is to state that, while the significance of the metabolites does not survive correction using FDR, there are trends of interest given the sample size in this study. I agree that this data is an important and useful addition to the field!

Reviewer #2: All of my previous concerns were addressed to my satisfaction and comments from the other reviewer and editor appear to be addressed as well. I appreciate the addition of the complete Spearman correlation analysis and feel that this helped round out the discussion of the increased energy harvest hypothesis.

I would suggest the following, very minor, changes before publication:

1. Lines 313 and 314, replace p<0.001 with exact p-values, if possible.

2. Line 398, consider replacing “authors” with “instigators”, only because I was immediately looking for authors of another study.

3. Line 422 unnecessarily uses “directly” twice; I would suggest removing the second usage.

4. For showing taxonomic abundance (Figure 3), I prefer stacked barcharts, but this is clearly a personal preference and given that the other reviewer specifically asked for a heatmap it works. Either is preferable to a pie chart.

7. PLOS authors have the option to publish the peer review history of their article (what does this mean?). If published, this will include your full peer review and any attached files.

Reviewer #1: **Yes: **Elliot S. Friedman

Reviewer #2: **Yes: **Laura Tipton

---

## [Author Response · Author response to Decision Letter 1]

6 Jan 2021

Response to reviewers.

Reviewer 1:

Comment:

I appreciate the author’s providing more information about diet. I still have some additional questions/concerns. Was this an outpatient or inpatient study? If inpatient I assume that diet was controlled by the staff (although explicit information would be helpful). However, if this was an outpatient study, how was dietary management implemented? Were meals provided by the study, or were meals prepared by subjects/subject’s families? Additionally, compliance with dietary guidance is a well-established issue in outpatient studies. Was this controlled at all? Finally, how long were subjects on the controlled diet prior to sampling? This would be important to determine whether the observed changes were a response to stimuli (diet) or a new steady-state composition/function of the microbiome and metabolites.

Response: 

We have elaborated on this information in the subsection Study Participants. It was an inpatient study (lines 111-112, 127), the diet was planned by a clinical dietitian, and the food prepared by the cafeteria kitchen of the Olivova Children’s Medical Institution and (lines 131-133). For further clarity, we added more details about the diet in lines 131-138 and added a statement regarding compliance (leftover food and/or additional intake of other items outside the provided meals could not be accounted for) in lines 139-140. The participants were not all sampled at the same time and sampling took place throughout the study period with the earliest collections taking place after at least one week of habituation to the prescribed diet (lines 142-144). Also, as noted in lines 137-138, the meals were based on recipes and foods typically eaten in a standard Czech school and home diet, thus, except for portion control, the meals did not represent an adjustment for most participants. 

Comment:

I appreciate the additional information regarding control subjects, thank you! It might be interesting to compare reference data sets (e.g., HMP, other pediatric studies) to determine whether the controls in this study are similar to others, but that does not need to be a requirement for publication.

Response: 

We did not add this to the manuscript, but we did take a look at some other control groups in studies that also divided youth into normal, overweight, and obese categories. The table appears in the "Response to Reviewers" letter that has been uploaded.

Comment:

I appreciate the additional information on participants, but the comment was in regard to the mock community used for sequencing controls.

Response:

We have now added the detailed composition of the mock community (lines 238-244).

Comment:

I appreciate the expanded discussion, thank you! I would recommend revising the wording of the statement “Despite consuming less kcal, the OW and OB groups still produced significantly more butyrate, which has been identified as the main energy supplier for colonic epithelial cells [8].” The measurement in this case is of butyrate in feces – which is butyrate produced by the microbiota but not consumed by the colonic epithelium.

Response:

We have revised the wording of this statement to be clear that there was a higher content of fecal butyrate, but not necessarily that more butyrate was produced (lines 398-401).

Comment:

This new figure conveys much additional information but is only at the phyla level. It would be helpful to include a genus level figure in addition to (not in replace of) the phyla level results. Additionally, the unsupervised clustering really shows the lack of segregation of microbial communities by group. I would suggest adding to the discussion the notion that while there are minimal differences in microbial community composition between groups, the metabolite data suggests that there may be differences in the metabolic activity of these microbes.

Response:

To depict data from another angle, we have now added a stacked bar charts figure at the genus level (Fig 4) showing the 20 most abundant taxa (by relative abundance), as well as the two genera which are significant (Escherichia and Tyzzerella subgroup 3). To keep the figure more readable and due to the very low relative abundances of most taxa, not all 83 identified genera were shown in the figure. We also added on the notion that gut microbiota functional activity, rather than composition, may have played a part in our observations (lines 479-487).

Comment:

Thank you for the response. I think that this is fine but should be mentioned in the manuscript. Perhaps a more appropriate way to address this is to state that, while the significance of the metabolites does not survive correction using FDR, there are trends of interest given the sample size in this study. I agree that this data is an important and useful addition to the field!

Response:

As suggested, we have now mentioned and addressed FDR in the manuscript (lines 220, 315-319).

Reviewer 2:

Comment:

Lines 313 and 314, replace p<0.001 with exact p-values, if possible.

Response:

We have not made this change due to the p-values being quite small (p = 2.220E-16 for acetate with butyrate, p = 3.363E-06 for arabinose with galactose), and to keep consistency with the number of decimal places for the other reported p values.

Comment:

Line 398, consider replacing “authors” with “instigators”, only because I was immediately looking for authors of another study.

Response: 

Edited as suggested (line 415).

Comment:

Line 422 unnecessarily uses “directly” twice; I would suggest removing the second usage.

Response:

Edited as suggested (line 439).

Comment:

For showing taxonomic abundance (Figure 3), I prefer stacked barcharts, but this is clearly a personal preference and given that the other reviewer specifically asked for a heatmap it works. Either is preferable to a pie chart.

Response:

Bar charts are a very good way to represent this information, and we have now added a stacked bar chart (Fig 4) based on genera % relative abundance to provide another angle to look at the taxonomic data.

---

## [Decision Letter · Decision Letter 2]

8 Feb 2021

Stool metabolome-microbiota evaluation among children and adolescents with obesity, overweight, and normal-weight using 1H NMR and 16S rRNA gene profiling

PONE-D-20-27287R2

Dear Dr. Havlík, ČZU v Praze,

We’re pleased to inform you that your manuscript has been judged scientifically suitable for publication and will be formally accepted for publication once it meets all outstanding technical requirements.

Kind regards,

Suzanne L. Ishaq, PhD

Academic Editor

PLOS ONE

Additional Editor Comments (optional):

Reviewers' comments:

Reviewer's Responses to Questions

**Comments to the Author**

1. If the authors have adequately addressed your comments raised in a previous round of review and you feel that this manuscript is now acceptable for publication, you may indicate that here to bypass the “Comments to the Author” section, enter your conflict of interest statement in the “Confidential to Editor” section, and submit your "Accept" recommendation.

Reviewer #1: All comments have been addressed

Reviewer #2: All comments have been addressed

2. Is the manuscript technically sound, and do the data support the conclusions?

Reviewer #1: Yes

Reviewer #2: Yes

3. Has the statistical analysis been performed appropriately and rigorously? 

Reviewer #1: Yes

Reviewer #2: Yes

4. Have the authors made all data underlying the findings in their manuscript fully available?

Reviewer #1: Yes

Reviewer #2: Yes

5. Is the manuscript presented in an intelligible fashion and written in standard English?

Reviewer #1: Yes

Reviewer #2: Yes

6. Review Comments to the Author

Reviewer #1: (No Response)

Reviewer #2: (No Response)

7. PLOS authors have the option to publish the peer review history of their article (what does this mean?). If published, this will include your full peer review and any attached files.

Reviewer #1: **Yes: **Elliot S. Friedman

Reviewer #2: **Yes: **Laura Tipton

---

## [Editor Report · Acceptance letter]

12 Mar 2021

PONE-D-20-27287R2 

Stool metabolome-microbiota evaluation among children and adolescents with obesity, overweight, and normal-weight using ^1^H NMR and 16S rRNA gene profiling 

Dear Dr. Havlík:

I'm pleased to inform you that your manuscript has been deemed suitable for publication in PLOS ONE. Congratulations! Your manuscript is now with our production department. 

Kind regards, 

on behalf of

Dr. Suzanne L. Ishaq 

Academic Editor

PLOS ONE